# Teaching Deep Learning, a boisterous ever-evolving field

**Alfredo Canziani** [1]

## Abstract

Machine and deep learning techniques are actively being developed with over 150 papers submitted daily to arXiv, each of which is introducing its own notation. To offer a course that reflects the latest developments of the field and illustrate them in a cohesive and consistent manner, one needs to systematically consume the literature, summarise and standardise it, implement working examples, and deliver a concise and consistent presentation of a given topic. This paper reports all the best practices developed by the author in his last decade of teaching experience.

## 1. Course curriculum creation

We shall consider the creation of a 14-week course, standard length for a single semester at my institution, comprising a 2-hour theory and 1-hour practical session a week.

It has proven effective to slice the content into smaller two-week-long units sharing a common theme, as shown in fig. 1. Not only do they provide a sense of uniformity, purpose, and clarity across the lessons, but they also help the student feel motivated and give them a sense of achievement when a given theme is completed.

In addition, the course is partitioned into two halves, $3\times$ 2-week foundational themes (with one or two additional weeks before the mid-term exam), and the remainder more advanced topics, often delivered by experts of the field, with the students focussing on the final Kaggle-like competition (Masters' students) or research project (PhD students). Holding a competition has proven very effective for evaluating the students uniformly by comparing their final scores on a given task. It also allows us to assess their ability to source relevant literature, open-source code, and techniques to maximise the final performance.

Finally, for each foundational theme in the first half of the semester, homework is assigned to have the students exten-

[1]Courant Institute of Mathematical Sciences, New York University. Correspondence to: Alfredo Canziani <canziani@nyu.edu>.

*Proceedings of the $2^{nd}$ Teaching in Machine Learning Workshop*, PMLR, 2021. Copyright 2021 by the author(s).

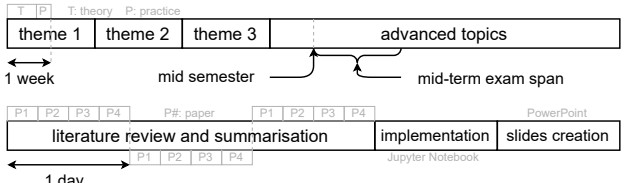

*Figure 1.* **Semester (top) and week (bottom) partitioning.** *Top:* each week comprises a 2 hour theoretical (T) and a 1 hour practical (P) session. Lessons are grouped in 2-week thematic blocks in the first half of the semester. *Bottom:* for a 3 day period, one may read and summarise 4 papers each day, use one day for the numerical implementation, and the remaining day for crafting the slides.

sively familiarise themselves with both mathematical and programming notions. Such assignments are intended to be intensive, geared toward students who are new to coding or need to review the fundamentals of linear algebra, calculus, or probability. Machine and deep learning papers and source codes publicly available demand fluency in both skills, which can be acquired only through exhaustive diligent practice.

## 2. Creating a single lecture

The process behind the creation of a single lecture has multiple stages: (1) literature review [3 days, 4 papers a day]; (2) numerical implementation [1 day]; (3) deck slides creation [1 day]. This time allocation is shown in fig. 1.

### 2.1. Literature review

We shall assume we lack specific knowledge of a given novel topic (not yet covered in textbooks) and need to read the literature to learn about the details. It is usually sufficient to cover a dozen papers, starting with the "original" one and following with the most cited follow-ups found by using websites like Semantic Scholar and Connected Papers.

One should summarise each paper content [2 hour budget] using LaTeX capable editors — such as Typora, Notion, HackMD (for Markdown solutions, which are more forgiving and faster to interact with[1]), or LyX (for higher quality

---

[1]Think about Markdown as an interpreted language and pure

outputs but slower interaction and less reusability in the next phase, described in section 2.2) — and adhering to the following template. **What:** one sentence describing **what** the article is about. **Why:** one sentence about **why** it is necessary. **How:** this is the main summary of the article, starting with the main figure(s) of the paper followed by a list of itemised logical steps illustrating the math accompanied by some telegraphically styled text. **And:** major takeaways and personal notes about the paper.

This structured phase is *essential* to gather some understanding across recent contributions to a specific area of research and allows to collect equations that will be helpful at deck slides creation time.

## 2.2. Numerical implementation

Once some familiarity with the topic has been acquired, we can work towards the implementation of a numerical example leveraging the concepts we just have learnt about [1 day]. More precisely, I draft the code from scratch in a Jupyter Notebook (Pérez & Granger, 2007) using Python (Van Rossum & Drake Jr, 1995), PyTorch (Paszke *et al.*, 2019), and Matplotlib (Hunter, 2007), intermingled by some sparse comments written in Markdown and LaTeX (which can be borrowed from the summaries previously made).

Depending on the novelty of a given topic, this stage is often very hard. After sourcing and processing information from a large pool of alternatives, we now need to consolidate it into a working example, which will provide the right amount of understanding and confidence for explaining the inner workings to others.

## 2.3. Crafting the slides

The objective is to communicate a new topic in the most effective manner. Since you know your students and their prior knowledge, you can focus on establishing a standard practice where the slides have minimal text beyond equations and the majority of the information is transmitted orally. A slide should **not** be used for sake of the presenter as a script of what should be said but should instead convey **visual** information to the listener: block diagrams, animations, or mathematical symbols (entities) and few equations (relations). Moreover, to guide the observer's attention (and the speaker's dialogue), fading in one element at a time provides the best pacing mechanism that allows dynamic sustained content delivery.

### 2.3.1. TOOLS

There are several alternatives for designing the slides; my personal choice is to use the ubiquitous Microsoft Power-

Point with a black background,[2] and a consistent 12-colour semantically used colour-scheme across **all** tools, as shown in figs. 2 and 3. Draw.io is used for drawing most of the network diagrams, which can be exported in SVG, and later ungrouped and modified within PowerPoint itself. *A few* mathematical symbols are introduced, using LaTeXiT, matching the diagrams colours. Charts and animations generated with Matplotlib[3] (Hunter, 2007), exported as PDF or MP4, can be and embedded in the deck, as shown in fig. 3. For each element added to the slides, one should assign a meaningful name, by accessing the `Selection Pane`, such that animations can be easily created, using the *fade in* option. Animations are fundamental to pace one's presentation and direct students' attention to one item at a time as they are appearing on screen, as noted in fig. 2.

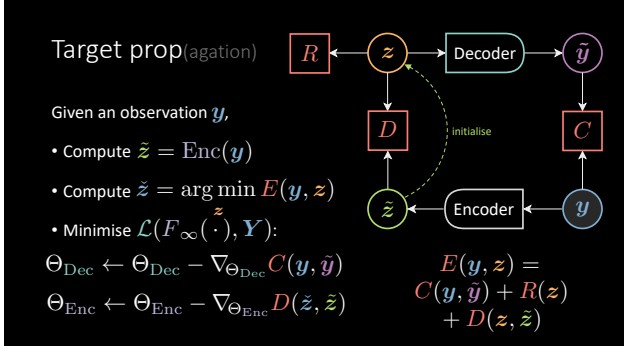

*Figure 2.* **Consistent semantic colour-coding.** *Notation*: green is used to represent a *hidden representation* throughout the course. The tilde $\sim$ on top of a variable represent an approximation and it means *circa* ('around' in Latin). Therefore, $\tilde{z}$ is an approximation of $z$, where the orange colour represents a *latent variable*. Similarly, $\hat{y}$ is an approximation for $y$. The blue (low-temperature) colour represents something with low energy. The check $\vee$ indicates the variable is the output of a minimisation process. In this slide there are three major elements: (1) top-right connection diagram; (2) algorithm on the left; (3) equation on the bottom-right. The algorithm and diagram appear one element at a time, synchronously. The equation for $E(y, z)$ appears after $E$ is used in the algorithm to compute $\check{z}$.

### 2.3.2. PRESENTATION PLAN AND STYLE

Given that we know what audience we tailor our presentation to, we shall solely focus on the length. Student focus decays after 15–20 minutes (Stuart & Rutherford, 1978) therefore, for a 1-hour lecture we will need 2–3 events to boost and replenish the attention. Such events need to be modestly disruptive, possibly involve emotions (*e.g.* sur-

---

LaTeX as a compiled one.

[2]Because blank screens are black and information is added by using light, as opposed to blank papers, which are white and information is added by subtracting light.

[3]Enabling LaTeX with `matplotlib.pyplot.rc('text', usetex=True)`.

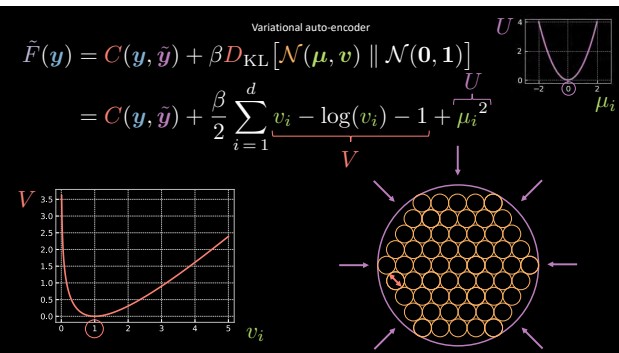

Figure 3. **Consistently coloured charts.** *E.g.* green is still used to represent a *hidden representation*, violet is used for $U$, the chart $U$ over $\mu_i$, and the radial inner force in the illustration.

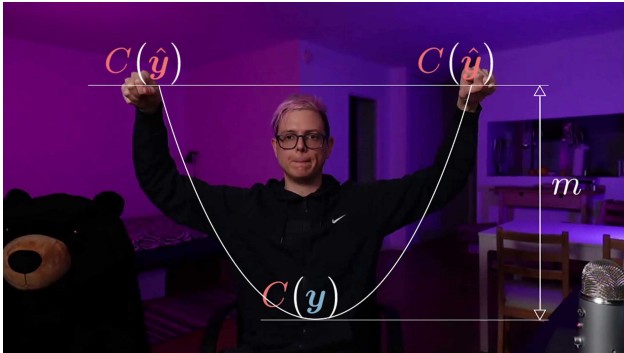

Figure 4. **Descriptive dynamic hand movements.** *Action*: lifted both hands to height $m$. *Animation*: both $C(\hat{y})$ terms follow the hands; the horizontal white line bends into a parabola. Precedently, $C(y)$ had been pushed down. *Notation*: the hat $\wedge$ and colour red (high-temperature) imply that the cost of $\hat{y}$ needs to be high. The colour blue (low-temperature) implies that $y$'s cost has to be low.

prise) and reactions (*e.g.* laughter). A given topic needs to be introduced by **why** we need it (to motivate the students and catch their attention) and **what** it is addressing (to clearly identify the subject) *before* attempting to explain **how** it works and comes together **and** its consequences and takeaways (notice similarities with section 2.1's summarisation).

Given that *clarity* is of fundamental importance, the 3 rules of 3∗ should be considered: (1) no more than 3 concepts per slides; (2) font size of 30 pt should be used; (3) between 30 seconds to 3 minutes per slide. If text is present, use bullet points, telegraphic text style (write out the full sentence, remove any unnecessary words), sans-serif font family, and **bold**, *italic*, and colours to highlight key concepts.

When possible, harder concepts should be accompanied by one or more physical analogies, to aid comprehension and allow the students to reason and ponder. Descriptive hand movements are also an effective and cheap tool to display dynamic interactions between moving parts. In case the lectures are recorded, one can later draw animations on top of such hand gestures to further aid understanding, as shown in fig. 4.

Finally, *everything* should be rehearsed several times, from making sure the animations play in the correct order and unveil the narrative logically, to the analogies and gestures.

## 3. Refining a lecture

Even if we followed all the steps proposed in section 2.3, students will ask to clarify logical gaps in the exposition of the topic. Such feedback is essential to make the lecture understandable and to improve it for the following semesters when new feedback will be provided. Such a cycle will smooth out with a few iterations. Faster convergence can be achieved by presenting your lecture in a group meeting.

It is also worth leveraging the Twitter's online scientific community to obtain feedback on more obscure topics perspective. The lack of a human component will allow you to collect unfiltered comments that address any shortcomings you may have overlooked. Moreover, one can query their virtual audience with quizzes and polls, and observe which learning strategy is of minimal resistance / lowest energy.

## 4. A note on accessibility and inclusion

Not everyone can obtain the highest amount of information from oral lectures held live and in spoken English. Therefore, redundancy in terms of available media ensures larger dissemination of knowledge.

Recording the lectures, even for internal use only, has proven very effective, especially if one may need to miss a day of school. Although the live student-teacher interaction would be missing, content can be still be consumed. Moreover, automatic captioning can facilitate those who may not be fluent in English or are hard-of-hearing. Closed captioning and captions translations can be obtained by leveraging the online community if the recordings are posted online and until recently[4] YouTube was permitting crowdsourcing such contributions. Furthermore, if one decides to pursue this option, it is helpful to add a table of contents to make the video searchable and more structured, for a more effective learning experience.

Lectures summaries can be assigned as voluntary homework, resulting in shared lecture notes, which are convenient to be used as reviewing material for the exams, since the slides

---

[4]The service has been discontinued in September 2020 to address spam / abuse problems (YouTube Help).

do not contain the narrative. Once again, if such notes are made publicly available, then translations can be obtained through open-source collaboration. Besides the utility in terms of consumption of alternative formats, summarising the content of the lectures has been an activity that several students enjoyed, letting them consolidate the knowledge of notions delivered in class.

# 5. Remote teaching

In case in-person teaching is not a viable solution, one needs to deliver their lectures remotely, making sure a clear and stable communication system is established between teacher and students. In this section, we will assume a symmetrical broadband high-speed Internet access is available to the instructor.

## 5.1. Hardware equipment

Having the best audio quality one can afford is of paramount importance, since students' concentration will be completely lost if the speech is unclear. Common hardware choices are USB microphones used by amateur podcasters and vloggers, such as the Blue Yeti or analogous condenser solutions. Any source of audible noise one has control over has to be removed, such as phone and computer notifications, air conditioning, closing open windows, *etc*...

The second most important aspect is an appropriate source of light for illuminating the instructor's face, helping separate the subject from the background. Virtual eye contact and face micro-expressions are broadly used to communicate nonverbal cues, emphasising core concepts, and sharing emotions with the audience. A strong connection with the spectator allows a smoother delivery of the lecture. An affordable dimmable LED or fluorescences softbox is sufficient for the task. When setting up the environment, one needs to make sure no light sources (*e.g.* windows, lamps) are present inside the recording frame, the background has a lower brightness than the foreground, and that the camera is set at eye level, with the eyes appearing at ⅓ from the top. Furthermore, dressing in dark solid colours will help the spectator focus on the instructor's face and hands, as shown in fig. 4. Alternatively, one may choose to do simply the opposite, and have lighter background and clothing, if this improves the contrast.

Finally, depending on the budget at disposal, one may want to invest in a mirrorless or compact camera with a clean HDMI output stream and with a 25 mm sensor or larger, and a 35 mm lens or shorter with an f/2.0 aperture or wider (*e.g.* the Sony ZV-1, Canon PowerShot G7 X Mark III). The HDMI output can be routed to a computer through a capture card.

## 5.2. Delivering a remote lecture

Interaction with the students is a requisite to be able to communicate effectively, regardless of whether in-person or remotely. A feedback loop, between teacher and students, needs to be established to sustain engaging learning dynamics. While in class this may be achieved by gauging students' attention through eye contact, online we need to prompt the students for "reactions" and / or to answer quick contextual questions to probe their awareness of the current train of thoughts. Conveniently, one can leverage instant messaging and virtual reactions available in videoconferencing software like Zoom. Students have displayed less timidity in formulating their questions in a textual format, and reply to each other in the text to clarify the content presented, displaying a stronger sense of community.

Although it may be initially hard, one should lecture while looking at the camera objective instead of the screen, unless digital eyesight adjustment is used, to recreate a virtual eye contact with the audience. Having a previewing monitor (software or hardware) is very convenient to measure the full extent of the area we have at disposal for hand gesturing. In addition, the mirror image helps the lecturer retain a human component when all other participants have their video feed disabled.

# 6. Privacy consideration

In order not to record students' appearance, remote environment, and voices, it is recommended to have them join without input audio and video. Furthermore, using text as a question formulation medium, the instructor can integrate the answers with more convenient timing, avoiding explanation flow interruptions.

# 7. Video editing and special effects

If one decides to post the video recordings online, some video editing may be required, using software like Adobe Premiere. Depending on the resolution of the footage, one may be able to export the composition in HD ($1280 \times 720\,\text{px}^2$) or FHD ($1920 \times 1080\,\text{px}^2$). The height of the camera feed should not be lesser than ½ of the vertical resolution, since facial expressions need to be easily visible. Videos played throughout the presentation have to be replaced with the original footage to prevent quality degradation and synchronisation issues. Additionally, one may augment the hand gestures with superimposed keyframed animated drawings, using software like Adobe AfterEffects and a graphic tablet, as shown in fig. 4.

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

YouTube Help. Turn on & manage community contributions. URL support.google.com/youtube/answer/6052538/.

Zoom. Video conferencing, cloud phone, webinars, chat, virtual events. URL zoom.us/.
