# OpenReview forum: "Teaching Deep Learning, a boisterous ever-evolving field"
_ecmlpkdd.org/ECMLPKDD/2021/Workshop/TeachML — TeachML 2021_

### Official Review · Reviewer_jeD2 · 2021-07-10
**Tips for teaching machine and deep learning**

**Rating:** 7
**Confidence:** 4

**Review:**

In this paper, the author presents the lessons learned while teaching machine learning and deep learning courses. The best part of the paper is the wide variety of tips that are provided by the author ranging from the setup to record lectures to how to create a single lecture. Those suggestions are not only useful for machine learning or deep learning topics, but also for any computer science subject.

The only critique that I have about the paper is that it is too focused on the teaching part, and the author does not discuss how the practical sessions are organised, and how much extra time are the students supposed to devote to the course. I would like to hear more about this.

---

### Decision · Program_Chairs · 2021-07-23

**Decision:**

Accept

**Comment:**

Congratulations! Your paper has been accepted. The reviewer and the PCs agree that the paper is well written. The PCs do agree that this paper focuses heavily on generic teaching tools that would benefit teaching any discipline.

Camera-ready version is due August 18, 2021. As you prepare the camera ready version, please take the reviewer's comments into consideration. The PCs recommend adding more specifics about teaching deep learning in particular, as the title of the paper suggests.

We look forward to your participation at the workshop on September 13, 2021. We invite you also to join us for the satellite event on September 08, 2021. Schedules for both the workshop and the satellite event will be forthcoming.